# Modeling of Hexavalent Chromium Removal with Hydrophobically Modified Cellulose Nanofibers

**DOI:** 10.3390/polym14163425

**Published:** 2022-08-22

**Authors:** Francisco de Borja Ojembarrena, Jose Luis Sánchez-Salvador, Sergio Mateo, Ana Balea, Angeles Blanco, Noemí Merayo, Carlos Negro

**Affiliations:** 1Department of Chemical Engineering and Materials, Complutense University of Madrid, Avda. Complutense s/n, 28040 Madrid, Spain; 2Department of Mechanical, Chemical and Industrial Design Engineering, ETSIDI, Universidad Politécnica de Madrid, Ronda de Valencia 3, 28012 Madrid, Spain

**Keywords:** wastewater treatment, adsorption, nanocellulose, cellulose nanofibers, hexavalent chromium, hydrophobization process

## Abstract

Cellulose nanofibers (CNF) are sustainable nanomaterials, obtained by the mechanical disintegration of cellulose, whose properties make them an interesting adsorbent material due to their high specific area and active groups. CNF are easily functionalized to optimize the performance for different uses. The hypothesis of this work is that hydrophobization can be used to improve their ability as adsorbents. Therefore, hydrophobic CNF was applied to adsorb hexavalent chromium from wastewater. CNF was synthetized by TEMPO-mediated oxidation, followed by mechanical disintegration. Hydrophobization was performed using methyl trimetoxysilane (MTMS) as a hydrophobic coating agent. The adsorption treatment of hexavalent chromium with hydrophobic CNF was optimized by studying the influence of contact time, MTMS dosage (0–3 mmol·g^−1^ CNF), initial pH of the wastewater (3–9), initial chromium concentration (0.10–50 mg·L^−1^), and adsorbent dosage (250–1000 mg CNF·L^−1^). Furthermore, the corresponding adsorption mechanism was identified. Complete adsorption of hexavalent chromium was achieved with CNF hydrophobized with 1.5 mmol MTMS·g^−1^ CNF with the faster adsorption kinetic, which proved the initial hypothesis that hydrophobic CNF improves the adsorption capacity of hydrophilic CNF. The optimal adsorption conditions were pH 3 and the adsorbent dosage was over 500 mg·L^−1^. The maximum removal was found for the initial concentrations of hexavalent chromium below 1 mg·L^−1^ and a maximum adsorption capacity of 70.38 mg·g^−1^ was achieved. The kinetic study revealed that pseudo-second order kinetics was the best fitting model at a low concentration while the intraparticle diffusion model fit better for higher concentrations, describing a multi-step mechanism of hexavalent chromium onto the adsorbent surface. The Freundlich isotherm was the best adjustment model.

## 1. Introduction

Chromium is a harmful water pollutant. Several environmental effects have been associated with the presence of trivalent and hexavalent chromium in natural water bodies. Trivalent chromium shows lower toxicity due to its low cell permeability, whereas hexavalent chromium presents acute toxicity to many species, causing carcinogenicity and mutagenicity [1,2,3], aside from being neurotoxic [4]. Furthermore, Cr(VI) is highly toxic and one of the most common environmental contaminants and is non-easily biodegradable in nature, thus staying in the environment for a long time, polluting the soil and water, with the subsequent health risks to humans and wildlife [5]. Several industries generate hexavalent chromium in their effluents including tanneries, electroplating, hardware, textile, cement, or mining industries [3,6]. The limit of discharge of this contaminant varies depending on the country, but most of European countries have low limits in water, between 0.05 and 0.1 mg·L^−1^ [7]. 

The most relevant hexavalent chromium removal treatments include adsorption, ion exchange, membrane filtration, electrochemical treatments, coagulation–flocculation, chemical precipitation, biological treatments, and photocatalysis [6,8,9,10]. Among them, adsorption is the most widely used technology due to the low operational costs, ease of operation, low environmental impact associated with the process, and its effectiveness in removing trace levels of chromium [11]. Some of these adsorption treatments are based on microbial bioadsorption [12], since some microbes are able to reduce hexavalent chromium to trivalent chromium; as the latter is much less harmful, their excretion by microbes is not an issue [5]. Common adsorbents such as activated carbons or zeolites have been successfully applied, but there is a need for new materials from renewable sources, with better biodegradability and whose raw materials have less environmental impact. Adsorbents coming from renewable sources include those obtained from waste phytomass, which could be valorized to adsorb hexavalent chromium, achieving 87.2% removal, with the possibility of regeneration for at least four cycle runs [13]. Agro-based biomass has also been used as a bioadsorbent, which contributes to significant sustainable waste management, with an adsorption capacity of more than 10 mg/g [12]. Other waste materials used to produce adsorbents are fishbone waste to produce hydroxyapatite, which successfully removed Ni^2+^, Cu^2+^, and Zn^2+^ (more than 95% removal at the optimum conditions) [14]. Furthermore, nanomaterials can be used as adsorbents such as the polyacrylonitrile nanofiber membrane modified with bovine serum albumin used to remove Ca^2+^ from process streams, achieving removal efficiencies of about 62% [15]. Other membrane structure materials used as adsorbents are, for example, cellulose acetate-based membranes with glass nanoparticles for CO_2_ separation [16]. The surfaces of the nanomaterial adsorbents are usually functionalized or chemically modified to improve Cr(VI) adsorption such as the attachment of active binding sites [17].

As cellulose is the most abundant biopolymer on Earth, and it is an inexpensive and renewable material, it is an excellent option as a raw material to produce new adsorbents [18,19] as well as its derivates such as sulfate cellulose [20].

Nanocelluloses are cellulosic materials with at least one dimension in the nanometer scale. There are different types of nanocelluloses including bacterial cellulose (BC), cellulose nanocrystals (CNC), and cellulose nanofibers (CNF). These materials show good adsorptive properties due to their high specific area and the presence of active groups such as carboxyl groups over their surface [21]. These nanomaterials have been proven as excellent adsorbents of different heavy metals such as cadmium, nickel, copper, lead, arsenic, iron, silver, cobalt, or mercury, thanks to their ease of functionalization and application, which enhances the adaptation to each heavy metal [22,23]. Celluloses are commonly anionic charged, caused by the pulping and bleaching processes [24]. Most of the heavy metals are cationic species, facilitating the direct application of nanocelluloses without any surface modification to obtain both a high removal yield and high adsorption capacity as well as fast adsorption rates. Khoo et al. [25] confirmed that there was a large affinity of CNC for Cu^2+^, Cd^2+^, Ni^2+^, Pb^2+^, Zn^2+^, Fe^3+^, and Ag^+^ adsorption, reached with none or low modifications. Compared to these cationic heavy metals, the adsorption capacity of CNC for anionic As(V) was reduced by an order of magnitude. The same trend was also observed by Liu et al. [26], who applied both untreated CNF and 2,2,6,6-tetramethylpiperidin-1-yl-oxyl (TEMPO)-mediated oxidized CNF to a list of 15 heavy metals, with hexavalent chromium the only anionic species. The authors found a maximum adsorption capacity of 87.5 mg·g^−1^ using TEMPO-oxidized CNF-PAN membranes to adsorb Cr(VI), which was lower than the adsorption capacity reached when cationic Pb(II) was adsorbed. 

The adsorption of anionic hexavalent chromium species with cellulosic materials usually requires other strategies and one of the most common options is surface modification. These surface modifications often include a chemical reaction, which forces a change in the surface groups present in the cellulose. Cationization, oxidation, esterification, alkaline treatment, or halogenation are some of the most typical treatments [25]. These modifications have demonstrated their applicability, and some of them such as dialdehyde oxidation or the cationization of celluloses have reached the large yields and adsorption capacities of hexavalent chromium [27,28]. Nevertheless, they usually imply a purification process that needs washing steps and the loss of reagents after treatment.

Another option could be the coating of the nanocelluloses. In this case, the nanocelluloses undergo a reaction where the coating reagent is completely mixed with the nanocellulose in suspension and the nanocellulose surface becomes covered by the coating agent. As this chemical is directly added to the final nanocellulose suspension, there is no need for separation after the reaction, avoiding the loss of material. Among the possible hydrophobization reactions, silanization becomes a great option because of the maintenance of large specific area of the fibrils and the enhanced hydrophobicity of cellulose fibers [29]. This fact allows for subsequent advanced treatments such as lyophilization to obtain aerogels [30]. Other silanized materials such as graphene oxide have been successfully applied to hexavalent chromium adsorption [31,32], but little information has been developed on the application of silanized cellulose-based materials to this contaminant. The closest approach is by Jamroz et al. [33], who indicated that amino-silanized celluloses could become an option for the ultra-trace determination of hexavalent chromium in water, which is not a direct wastewater treatment.

The state-of-the-art shows that there is a lack of studies about the use of nanocellulose to remove Cr(VI) from water streams, and particularly, the efficiency of hydrophobic nanocelluloses is not known. Furthermore, the kinetic and adsorptive behavior of these modified materials have not been well-studied yet. On the other hand, in this study, the applied hydrophobization method was based on a coating process to reduce product losses.

Therefore, in this study, the main objective was the synthesis and characterization of CNF hydrogels modified with methyl trimetoxysilane (MTMS) as a hydrophobic coating agent and their application to Cr(VI) adsorption from wastewater. The novelty of this approach is related to the application of hydrophobic CNF, hydrophobized by a coating method, to improve the adsorption performance of cellulosic nanofibers. Furthermore, the proposed adsorbent material application was optimized for the removal of Cr(VI) by performing a set of experiments on a batch system to evaluate the effect of system variables, MTMS dosage, pH value, initial chromium concentration, adsorbent dosage, and contact time. The kinetic and isotherm data were modeled to increase the knowledge related to the process, making easier its future application. Furthermore, different parameters have been obtained from a wide variety of models, which are the key factors in understanding the way hexavalent chromium interacts with the surface of the tested nanocelluloses.

## 2. Materials and Methods

### 2.1. Materials

Bleached pine kraft pulp was supplied by Arauco (Chile) and used as a raw material to synthetize CNF hydrogels. All of the experiments were performed using Milli-Q quality water. The chemicals used for the production and characterization of nanocelluloses were hydroxylamine hydrochloride, silver nitrate, methyl trimetoxysilane (MTMS), and 2,2,6,6-tetramethylpiperidin-1-yl-oxyl (TEMPO) supplied by Sigma Aldrich; sodium bromide, potassium chromate, sodium chloride, sodium hydroxide pellets, and sodium hypochlorite solution (10 *w*/*v*%) supplied by Panreac; and hydrochloric acid (37% *v*/*v*) and sulfuric acid (98% *v*/*v*) were supplied by Labkem. All of the chemicals used were of analytical grade.

To analyze hexavalent chromium, standard solutions were used. Hexavalent chromium standard solution (50 mg·L^−1^ as Cr(VI)) was used as a calibration standard for the spectrophotometric method. Analytical reagents for hexavalent chromium determination were purchased from Macherey Nagel (Dueren, Germany) following Standard Method 3500 Cr B. Poly(diallyldimethylammonium chloride) (PDADMAC) and polyethylenesulfonate (PesNA) solutions with a concentration of 0.00025 N were used as standard titration reagents during cationic and anionic demand determination. 

### 2.2. CNF Hydrogel Synthesis

Briefly, bleached pine kraft pulp was disintegrated at 30,000 revolutions and diluted at 1% of consistency. Then, TEMPO-mediated oxidation was carried out through the addition of 5 mmol of NaClO per gram of cellulose using TEMPO and NaBr at 0.1 mmol and 1 mmol per gram of cellulose, respectively [34,35,36,37]. The oxidized cellulose suspension, after a cleaning process, was then treated under three-step mechanical homogenization at 600 bar pressure. Finally, the homogenized suspension underwent surface modification with MTMS to perform the hydrophobization, as indicated by Zhang et al. [30].

### 2.3. CNF Characterization

The characterization of the CNF hydrogels differs from the typical physical-chemical characterization of adsorbents as it is a hydrogel and must be characterized as a nanocellulose suspension instead of the common solid characterization. This characterization included the determination of the consistency of the suspension as well as the anionic demand and transmittance of the CNF suspension at λ = 800 nm, as described by Balea et al. [38], and the zeta potential of cellulose nanofibers and the amount of carboxyl groups in the oxidized pulp suspension determined by conductimetric titration as described by Sanchez-Salvador et al. [39]. 

### 2.4. Experimental Procedure: Batch Adsorption of Hexavalent Chromium Solution

The experimental installation consists of a batch, stirred laboratory beakers of 250 mL filled with 100 mL of sample of synthetic wastewater placed on multiple position hot plates under temperature control. Different operating conditions were studied: contact time, pH, chromium concentration, and adsorbent dosage. In the case of the CNF hydrogel, the dosage of hydrophobic reagent MTMS in the synthesis route was also optimized regarding adsorption efficiency. 

During the kinetic experiments, the adsorption contact times were considered, taking samples at 1, 5, 15, 30, 60 min, 2, 3, 4, 24, and 48 h. The samples were filtered by cellulose acetate syringe filters (pore size 0.45 μm) to separate the insoluble fraction of the adsorbent from the soluble fraction as this material did not interact or adsorb chromium while passing through. Each experiment was considered finished (equilibrium time) when the pollutant level remained constant in 2–3 samples in a row. The hexavalent chromium concentration was determined by using a spectrophotometer calibrated with diluted solutions from a standard of chromium of 50 mg·L^−1^ between 0.02 and 0.50 mg·L^−1^ and the measurement was performed at the peak wavelength λ = 540 nm, as indicated in Standard Method 3500 Cr B [40]. Synthetic hexavalent chromium solution samples were diluted when the concentration exceeded the top of the calibration curve. 

The pH effect was evaluated from acidic (pH 3) to neutral and alkaline (pH 9) conditions through the addition of HCl and NaOH, both at 0.1 mol·L^−1^. Hexavalent chromium concentration in the kinetic and isotherm studies varied from 0.1 to 50 mg·L^−1^. Hydrophobized CNF hydrogels were tested in a dosage range 250–1000 mg CNF·L^−1^. The dosage of the hydrophobic reagent MTMS during CNF hydrogel synthesis was varied from 0 to 5 mmol MTMS·g^−1^ CNF hydrogel, according to the indications from Zhang et al. [30]. Each test was repeated three times.

### 2.5. Isotherm and Kinetic Studies 

The conversion of the hexavalent chromium concentration into adsorption capacity values was determined. Efficiency of the Cr(VI) removal and adsorption capacity was calculated according to Equations (1) and (2) [41].
(1)$$\%\;\mathrm{Cr}\left(\mathrm{VI}\right)\;\mathrm{removal}=\frac{\left(C_{0}-C_{t}\right)}{C_{o}}\cdot 100\;$$
(2)$$q_{\mathrm{Cr}\left(\mathrm{VI}\right)}\left[\frac{\mathrm{mg}\;\mathrm{Cr}\left(\mathrm{VI}\right)\mathrm{adsorbed}}{g\;\mathrm{CNF}}\right]=\frac{\left(C_{0}-C_{t}\right)}{m_{\mathrm{ads}}}\cdot V_{\mathrm{water}}$$
where C_0_ and C_t_ represent the initial and equilibrium concentrations of pollutants in solution, respectively (mg·L^−1^); V_water_ is the volume (L) of the solution; m_ads_ is the adsorbent mass (g CNF). The adsorbent mass (m_ads_) can be calculated by the following Equation (3), which relates to the volume dose of the adsorbent and the consistency measured as indicated before.
(3)$$m_{\mathrm{ads}}\;\left[g\;\mathrm{CNF}\right]={\;V}_{\mathrm{ads}}\cdot \mathrm{Consistency}\;$$
where V_ads_ is the volume of adsorbent and the consistency is the dried mass of adsorbent at 60 °C [42]. These compiled data were subsequently analyzed through different kinetic models.

The experimental kinetic and isotherm data were fitted through different kinetic and isotherm models to identify the adsorption mechanisms of the hexavalent chromium onto the surface of the CNF. The selected kinetic and isotherm equations and their linearized forms are summarized in Table 1 and Table 2.

## 3. Results and Discussion

### 3.1. CNF Characterization

The cellulose-based physical-chemical characterization begins with the determination of the carboxyl group content on the oxidized cellulose before homogenization. Carboxylic acids created from primary hydroxyl groups on the cellulose leads to an increase in the repulsion between individual fibrils that form the cellulose fiber, which subsequently favors the homogenization process. Therefore, the number of carboxylic groups formed is indicative of the technical feasibility of the homogenization process, particularly to avoid clogging the device. The oxidized cellulose achieved 1.099 mmol COOH·g^−1^. This content was higher than the values reported by Patiño-Masó et al. [48] (0.75 mmol COOH·g^−1^), Lu et al. [49] (0.73 mmol COOH·g^−1^) and Balea et al. [38] (0.50 and 0.25 mmol COOH·g^−1^) using bleached kraft eucalyptus pulp, bleached bagasse, recycled newspaper, and corrugated container as raw materials under equivalent NaClO dosages, respectively. These differences are related to the quantity of impurities and lignin present in the source of cellulose, which has a strong influence on the amount of carboxylic groups formed for a specific amount of NaClO [38]. Therefore, the high number of carboxylic groups that formed was due to the reduced amount of impurities and lignin in the bleached pine kraft pulp used in this case, which agrees with the good performance of the oxidized cellulose during the homogenization process to obtain the CNF, and explains the low number of homogenization cycles (three). 

Once the CNF hydrogel was obtained by homogenization, it was characterized by measuring the zeta potential of the CNF suspension. Zeta potential is indicative of the stability and separation of individual cellulose nanofibers, negatively charged; therefore, as its value becomes more negative, the suspension has more stability due to the repulsion between nanofibers that avoid aggregation between them. The value of the zeta potential for this suspension was −41.33 mV, which revealed the negative charge on the nanocellulose surface and the stability of the CNF suspension. In the literature, zeta potential values below −30 mV are indicators of bilateral repulsion and colloidal stability [50,51]. Other authors have obtained similar results for nanocellulose from acid hydrolysis (−38.2 mV) and TEMPO-oxidized nanocellulose (−46.5 mV) caused by the sulfonate and carboxyl groups, respectively [52]. 

Aside from the zeta potential, the cationic demand of the CNF hydrogel was measured, resulting in 838.5 µeq·g^−1^. This value was higher than the one obtained by Balea et al. [38] for recycled fibers (200–600 µeq·g^−1^) and lower than those obtained by Patiño-Masó et al. [48] (1000 µeq·g^−1^) for bleached kraft pulp from eucalyptus under the same testing conditions. Cationic demand is indicative of the degree of defibrillation achieved by homogenization. As the cellulose surface is negatively charged, a higher amount of negatively charged groups in the suspension means that the specific surface of the cellulose material/nanomaterial was higher. Therefore, the cationic demand will be higher as the specific surface increases, which implies that a higher number of individual nanofibers was achieved during the homogenization process. The degree of nanofibrillation achieved was directly related to both the cellulose source and the cycles applied during the homogenization, which explains the higher value achieved regarding recycled fibers such as the one obtained with virgin fibers, as the amount of cellulose in virgin sources is higher than the cellulose in recycled ones.

To evaluate the amount of nanofibers in the CNF suspension, the degree of nanofibrillation in the CNF, the transmittance at 800 nm was determined [53]. The value of transmittance achieved was 95.6%, indicating that the synthetized CNF were highly nanofibrillated. As a comparison, Patiño-Masó et al. [48] obtained a 99.13% of nanofibrillation yield with a suspension of 88.0% of transmittance. This fact shows that most of the CNF can be considered disaggregated and separated in small individual nanofibers through this treatment. 

### 3.2. Kinetics of Cr(VI) Adsorption with Hydrophobic CNF

#### 3.2.1. Effect of MTMS Dosage

The effect of MTMS dosage, added to coat the surface of the CNF, on the hexavalent chromium adsorption was evaluated. Three MTMS dosages were tested (0, 1.5, and 3 mmol MTMS·g^−1^ cellulose). These adsorption batch experiments were evaluated at pH 3, 1000 mg·L^−1^ of adsorbent dosage, and 0.1 mg·L^−1^ of initial chromium concentration. Hexavalent chromium concentration in the water was measured along time, using the three different hydrophobic CNF used as adsorbents (Figure 1). The trends of chromium decreased for each dose of MTMS, indicating a variation in both the adsorption rate and the maximum adsorption capacity. Whereas the CNF hydrogel without the hydrophobic coating reached a maximum chromium removal of about 80%, after 75 h of contact time, both the MTMS doped CNF hydrogels showed a higher adsorption capacity as they adsorbed hexavalent chromium up to complete abatement, after 75 h of contact time. However, the lowest MTMS dosage applied to the CNF enhanced the adsorption rate (more than 90% of chromium was removed before 6 h of treatment) in comparison with the CNF without the MTMS coating and the CNF with the highest MTMS dosage. Other modified CNF (i.e., cationized [54], carboxylated [55,56], acid treated [57], and diethylenetriamine [58] modifications) achieved similar removal yields, over 90% of hexavalent chromium removal after 120 min of contact time using doses of the adsorbent between 0.3 and 3 g adsorbent·L^−1^ at acidic pH values from 1 to 5.5.

Following the calculation methodology described in Section 2.5, the adsorption capacity was determined and plotted in each experiment. Then, these data were fitted according to the different kinetic equations proposed in Table 1 (Figure 2).

The best fitting model in this case was the pseudo-second order kinetic equation, which corresponded to a saturation mechanism. The fitted kinetic parameters achieved through the evaluated models are supplied in the Appendix A (Table A1). The adjustment of the intraparticle diffusion model showed a multi-step adsorption process. The first linear adjustment of all experiments showed a high correlation coefficient and the intercept was close to the origin, meaning that internal diffusion is a rate-limiting step due to both the linearity and the number of steps, and the low boundary layer effect was considered according to the intercept [59]. Pseudo-second order kinetics and multi-step kinetic models were also indicated by Xu et al. [60], who used black wattle tannin-modified dialdehyde nanocellulose to adsorb hexavalent chromium. These authors associated this pseudo-second order fitting to the possibility of diffusion as a rate-limiting step.

Comparing the nonlinear adjustment of the pseudo-second order for each MTMS dosage applied to CNF, the maximum adsorption capacity corresponded to 0.30 mg Cr(VI)·g^−1^ CNF when applying the 1.5 mmol MTMS·g^−1^ CNF hydrogel (Figure 3), confirming the experimental results plotted in Figure 2. Non-modified CNF and hydrophobized CNF with 1.5 mmol MTMS·g^−1^ worked similarly, both reaching high adsorption capacities and fast saturation, while hydrophobized CNF with 3 mmol MTMS·g^−1^ showed slower adsorption, and saturation of the adsorbent was not found at 24 h of operation. The results of the adsorption capacity in the equilibrium and contact time to equilibrium were close to those found by other silanized cellulose applied by Jamroz et al. [33] to adsorb hexavalent chromium of 0.30 mg·g^−1^ at 300 min, respectively. However, these authors applied hydrophobic cellulose to measure ultra-traces of Cr(VI) in water. They also found that the pseudo-second order kinetic was the best fitting model. 

The slightly MTMS-coated CNF adsorbed chromium better than the naked CNF as the CNF surface groups were hidden by the coating layer, thus reducing the interaction of these groups with water. Furthermore, this technique prevented electrostatic repulsion between the negative CNF-surface charges and the hexavalent chromate ions. Nevertheless, excessive coating of MTMS over the surface of CNF supposed a mass transfer limitation, minimizing the adsorption rate. For these reasons, the selected MTMS dosage for the resting adsorption tests was 1.5 mmol MTMS·g^−1^. Compared to other silanized materials applied for hexavalent chromium removal, this amount of silanization agent was considerably lower. Around 40 mmol·g^−1^ of silanization reagents ((3-aminopropyl)trimethoxysilane, [3-(2-aminoethylamino)propyl]trimethoxysilane, and 3-[2-(2-aminoethylamino)ethylamino]propyl-trimethoxysilane) were added to coat the graphene oxide to adsorb hexavalent chromium from water [32].

#### 3.2.2. Effect of pH

The experimental evolution of the hexavalent chromium concentration along the batch adsorption tests by modifying the pH from pH 3 to pH 9 is shown in Figure 4. These adsorption tests were carried out by adding the best tested operating condition of MTMS dosage, 0.1 mg·L^−1^ of initial chromium concentration, and 1000 mg·L^−1^ of CNF dosage. The trend of the Cr(VI) removal rate was similar under neutral (pH 7) and alkali (pH 9) conditions, while the adsorption was faster under acidic (pH 3) conditions, with a total removal of 80% after 1 h of operation (Figure 4). The high adsorption capacity found at pH 3 was associated with the equilibrium changes of hexavalent chromium under acidic conditions. While divalent chromate is the predominant specie when pH >6, the monovalent specie is mainly present between pH 2 and 4. Therefore, the amount of adsorbate, hexavalent chromium, is doubled at acid pH because only one active site is required per anion. A similar result of pH optimization can be found with independence of the kind of adsorbent, as indicated by Owlad et al. [9] and Saha and Orvig [61]. It can be concluded that the pH effect is related to the adsorbate ionic forms, being the effect on the adsorbent negligible. After these experiments, the pH 3 condition was selected for the rest of the optimization process.

The obtained adsorption capacity data were evaluated through different kinetic models. The result of plotting each fitted kinetic to the pH 3 experimental data can be observed in Figure 5.

The adjustments demonstrate that the most representative model was the pseudo-second kinetic model (Figure 5). The fitted kinetic parameters obtained in the different pH experiments are shown in Table A2. The intraparticle model analysis showed a multi-step adsorption mechanism with independence in the pH, as seen in the MTMS optimization. The representation of the pseudo-second order kinetic adjustment to the experimental data from each experiment is seen in Figure 6. The maximum adsorption capacity was achieved while operating under pH 3 conditions, when 0.30 mg·g^−1^ of the adsorption capacity and 80% of the maximum removal was reached in 1 h. The operation under neutral and alkaline media was similar in adsorption rate, but with a slightly lower adsorption capacity when operating at alkaline conditions. 

#### 3.2.3. Effect of Adsorbent Dosage

Following the same experimental procedure of the previous optimization processes, the adsorbent dosage effect was also evaluated from 250 up to 1000 mg CNF·L^−1^ (Figure 7). The kinetic curves showed a clear tendency of increasing the adsorption rate as the dosage of CNF increased. Among the studied CNF dosages, the minimum one needed to obtain a 100% of adsorption of hexavalent chromium was 500 mg·L^−1^ whereas the lowest adsorbent dosage studied showed a fast saturation of the CNF and a total removal below 10% (Figure 7). The highest dosage studied, 1000 mg·L^−1^, achieved the complete removal of hexavalent chromium after 25 h of contact time, the same as the dosage of 500 mg·L^−1^. However, the higher the dosage, the faster the adsorption, as expected. In terms of the adsorption capacity, the optimum value was found while applying 500 mg·L^−1^, reaching the largest efficiency in chromium removal per gram of CNF. 

The quantity of hydrophobic CNF needed to reach the complete removal of hexavalent chromium is in most cases lower, compared to other cellulosic materials. Xu et al. [60] reached the maximum adsorption capacity of hexavalent chromium using 500 mg·L^−1^ with black wattle tannin-modified dialdehyde nanocellulose. Qiu et al. [62] and Huang et al. [58] achieved the complete depletion of hexavalent chromium employing 3000 mg·L^−1^ of polyethylenimine facilitated ethyl cellulose and diethylenetriamine-modified hydroxypropyl methylcellulose, respectively. In the study developed by Singh et al. [63], a dosage of 5000 mg·L^−1^ of aminated cellulose nanocrystals adsorbed 98.33% of hexavalent chromium. These results depict that hydrophobic CNF reached equivalent yields compared to other cellulosic materials, reducing the dosage by an order of magnitude. 

Compared to other kinds of adsorbents, the dosage of hydrophobic CNF can still be considered reduced. In the case of waste material adsorbents such as treated sawdust [64], pristine almond green hull [65], and *Hibiscus cannabinus* kenaf [66] were applied to treat hexavalent chromium under optimal conditions, 1600, 2000 and 3000 mg·L^−1^ were necessary, respectively. Other typical materials used to adsorb hexavalent chromium are both commercial and mango kernel-synthetized activated carbons [67,68], and chitosan microparticles and nanoparticles [69], but they require larger adsorbent dosages than the hydrophobic CNF. The optimal dosage needed for the activated carbons was 2000 mg·L^−1^, while 800 mg·L^−1^ was the minimum required dose of the chitosan microparticles and nanoparticles to reach hexavalent chromium removal.

The results of adjusting the selected kinetic models to a dosage of 500 mg·L^−1^ are shown in Figure 8. The optimal kinetic model corresponded to a pseudo-second order equation. All of the kinetic parameters obtained through the kinetic fittings from the dosage optimization experiments are shown in detail in Table A3. The results of the intraparticle diffusion model adjustments were similar for each dosage, corresponding to multi-step adsorption. 

Figure 9 represents the final pseudo-second order kinetic curves fitted to the experimental data from adsorbent dosage evaluation. The evolution observed revealed a maximum adsorption capacity of 0.58 mg·g^−1^ when 500 mg CNF·L^−1^ was dosed compared to 0.30 mg·g^−1^ in the case of 1000 mg CNF·L^−1^. However, the pseudo-second order kinetic constant was fourteen-times higher in the case of the highest dosage. The contrast between reaching fast-equilibrium with high adsorbent dosages and increasing the adsorption efficiency can be observed through the use of other nanomaterial adsorbents such as Fe_2_O_3_ nanoparticles [70]. This fact would suggest that the optimal dosage would depend on the objective between reaching the rapid total removal or maximizing the total capacity of the adsorbent. 

#### 3.2.4. Effect of Initial Chromium Concentration

Hexavalent chromium concentration was modified to study the adsorption capacity of CNF when the initial chromium concentration was between 0.1 and 5 mg·L^−1^ and when the initial chromium concentration drastically increased to the range from 10 to 50 mg·L^−1^ (Figure 10a,b, respectively). These kinetic experiments were developed by implementing the optimal tested MTMS dosage, the optimal tested pH, and the maximum adsorbent dosage (1000 mg CNF·L^−1^), thus prioritizing the process kinetic over the adsorption capacity. Furthermore, this CNF hydrogel dosage was selected to prevent the CNF from extremely fast saturation. 

All of the studied initial chromium concentrations showed similar trends, with an initial fast adsorption of chromium, a second stationary step, and a final slow adsorption step (Figure 10). This situation corresponded to a multilayer adsorption. At the beginning, the free active sites were easy to find, so a first layer was generated rapidly. Once the first layer was formed, there was an accumulation of anions in the CNF surface, which led to the minimization of both the driving force between the liquid and solid phases and the electrostatic force interaction. Therefore, the stationary and intermediate step is caused by the slow reduction of anionic hexavalent chromium to less toxic cationic trivalent chromium. This reduction of chromium through the oxidation of C–OH surface groups to carboxylic groups was studied deeply by Wang and Lee [19]. Afterward, the attachment of the second layer of hexavalent chromium onto the CNF surface begins once the surface gradually becomes positively charged and the attraction force of anionic hexavalent chromate from the wastewater to the cationic surface of the CNF overpasses the mass transfer limitations. From the results shown in Figure 10, the resulting kinetic curves of hydrophobic CNF adsorption when modifying the initial chromium concentration showed an increase in the adsorption capacity (from 0.30 to 70.38) as the initial concentration of chromium increased, coupled with a reduction in the adsorption efficiency (from 100% to 51%) as the initial chromium concentration increased. This fact demonstrates that both parameters, adsorption capacity and adsorption efficiency, showed opposite trends, in agreement with Pourfadakari et al. [57], as for a selected dosage, increasing the initial concentration involves both a lesser percentage of chromium removal and a higher driving force between the liquid and the solid surface. This is due to the difference in the concentrations, which enhances the efficient usage per mass unit of material, reaching a higher amount of adsorbed chromium per active site. These authors [57] also showed the change in the shape of the curves while increasing the initial concentration, where the presence of different steps became clearer as the concentration rose.

The achieved experimental data were fitted through the kinetic models applied previously, with the initial hexavalent chromium concentration of 25 mg Cr(VI)·L^−1^ (Figure 11). At this initial concentration, as it happened at 1 mg·L^−1^, the only model that allowed an adequate interpretation and simulation of the experimental results was the intraparticle diffusion model. Each kinetic parameter obtained by kinetic analysis for all the kinetic models can be checked in Table A4. The intraparticle diffusion model predicts a three-step adsorption mechanism including the first high adsorption rate step, a steady-state step, and the last slow adsorption step, confirming the trend seen in the experimental data. Evaluating the intraparticle adjustment of the first step in the experiments using an initial concentration of 5 mg·L^−1^ and above, the intercept is extremely high, in some cases, close to the maximum adsorption capacity. This fact indicates a strong effect of the boundary layer, which means that external diffusion limitation will play a major role in the overall adsorption rate [59]. 

The resulting curves of each initial chromium concentration experiment adjusted to the pseudo-second order kinetics (Figure 12a,b) showed the change in the mechanism of adsorption. At the lower initial concentration of hexavalent chromium, the adsorption mechanism was mainly saturation, whereas as the initial concentration increased, the multistep adsorption mechanism was more noticed. The final values of both equilibrium concentrations of chromium and the adsorption capacities of CNF achieved in these tests were used to plot the isotherm curves. The low correlation coefficients for the kinetic models were due to the assumption that all of the selected kinetic equations made, as they suppose the gradual increase in the adsorption capacity or continuous adsorption up to saturation of the adsorbent, but none of these models assumed a two-layer adsorption. Nevertheless, as it was previously pointed out, the kinetic data showed a third step in the adsorption process, corresponding to the adsorption at long contact times, longer than 20 h of batch adsorption. These long contact times are not feasible at an industrial scale; therefore, the selection of experimental data with shorter contact times (first and second steps) would enhance the adjustment of these models. 

There are a large variety of adsorbents to remove hexavalent chromium and many results related to how the contact between water and adsorbent is made, and to the different surface modifications, which affects the mechanism of adsorption. The hydrophobic CNF hydrogel showed the slowest equilibrium time regarding other adsorbent materials based on nanocellulose or on activated carbon (Table 3), but it reached the complete abatement of chromium and good adsorption capacity. Similar results of the maximum adsorption capacity in the equilibrium using polyacrylonitrile-modified CNF membranes were reported by Yang et al. [71] (87.5 mg·g^−1^). The equilibrium time of the hydrophobic CNF was similar to the one of another silanized cellulosic material applied by Jamroz et al. [33] (300 min) to detect ultra-trace concentrations of hexavalent chromium in water, suggesting that silanization processes lead to larger contact times than other kinds of adsorbents (Table 4) and could be associated with relevant mass transfer limitations due to the silane reaction with celluloses. The applied dosage was in the order of magnitude of other cellulosic adsorbents such as polypyrrole-bacterial CNF and polyaniline-functionalized CNF and was lower than that of activated carbons [72,73,74,75,76]. This comparison suggests that the hydrophobized CNF hydrogel is an efficient material for hexavalent chromium adsorption from wastewater compared to other adsorbents including other nanocellulosic materials.

### 3.3. Isotherm Analysis

The isotherm curves were graphed with the equilibrium data of the chromium concentration. The equilibrium data of the maximum adsorption capacities and final concentration at the equilibrium stage of hexavalent chromium treated with the hydrophobic CNF hydrogel were analyzed and adjusted to different isotherm models. The isotherm data showed an exponential increase trend instead of a saturation trend (Figure 13). According to the classification established by McCabe, Smith, and Harriott [78], this concave upward curve involves a large effect of mass transfer limitations and indicates unfavorable adsorption. This mass transfer limitation can be associated with the coating of MTMS on the surface of the material, which implies more tortuosity to reach the active sites. The parameters achieved from the adjustment of the different isotherm models are summarized in Table 3. The optimal correlation parameters are found when the Freundlich isotherm is applied to experimental data, which means the distribution of the energy through an exponential equation, and considering the heterogeneous dispersion of the active sites, happens over the adsorbent surface [79]. This mechanism supposes a multilayer adsorption [80], which fits the trend seen during the adsorption kinetics in Section 3.2.

**Figure 13 polymers-14-03425-f013:**
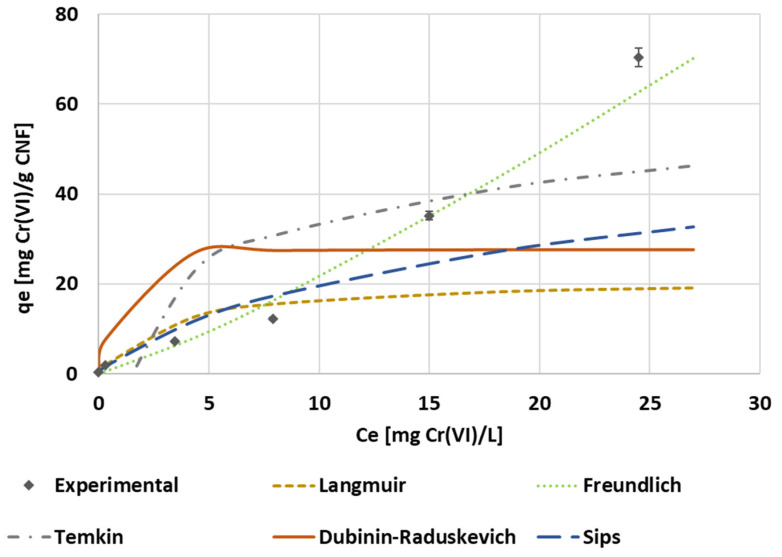
The isotherm experimental data and isotherm model adjustment of the Langmuir, Freundlich, Temkin, Dubinin–Raduskevich, and Sips equations.

**Table 4 polymers-14-03425-t004:** The results of the isotherm model adjustment to the adsorption equilibrium data of hexavalent chromium on a CNF hydrogel.

Model	Parameters	Values
Langmuir	Isotherm parameters	k_L_ [L·mg^−1^] = 21.26 q_e_ [mg·g^−1^] = 0.3417 R_L_ (C_0_ = 0.1 mg·L^−1^) [-] = 0.9670 R_L_ (C_0_ = 50 mg·L^−1^) [-] = 5.53·10^−2^
	Correlation parameters	R^2^ = 0.7420 RSS = 2949.55
Freundlich	Isotherm parameters	k_F_ [mg^(1−1/n)^-L^(1/n)^·g^−1^] = 1.3914 n_F_ [-] = 0.8404
	Correlation parameters	R^2^ = 0.9902 RSS = 108.01
Dubinin–Raduskevich	Isotherm parameters	B_DR_ [mol^2^·J^−2^] = 9.93·10^−8^ q_max_ [mg·g^−1^] = 27.72
	Thermodynamic parameters	E_DR_ [J·mol^−1^] = 2243.50
	Correlation parameters	R^2^ = 0.5754 RSS = 2542.39
Temkin	Isotherm parameters	B_T_ [J·mol^−1^] = 12.83 b_T_ [-J·mol^−1^] = 188.08 A_T_ [L·g^−1^] = 1.3759
	Correlation parameters	R^2^ = 0.7481 RSS = 1415.93
Sips	Isotherm parameters	n_S_ [-] = 1.2442 k_S_ [L^(1/nS)·^mol^-(1/nS)^] = 6.16·10^−2^
	Correlation parameters	R^2^ = 0.9023 RSS = 1529.83

The value of the parameter n_F_ <1 indicates an unfavorable process as well as the small bond adsorbate–adsorbent compared to a favorable process [81]. A similar value of n_F_ was reported by Dawodu et al. [82] for hexavalent chromium adsorption onto the seed coat biomass, showing a cooperative adsorption between Cr(VI) and adsorbent surface.

As expected by the shape of the isotherm, the selected models, which supposed the saturation of the adsorbent such as Langmuir, Sips, or Dubinin–Raduskevich, showed low correlation coefficients and a high residual sum of squares. These could represent the initial part of the curve, but failed the clear concave shape adjustment in the end. For this reason, these models did not offer a good estimation of the maximum adsorption capacity reached by the hydrophobic CNF. 

The Langmuir’s separation factor values calculated for the minimum and maximum initial concentrations were in the interval of 0 < R_L_ < 1, but close to the upper and lower limits of the interval, respectively. The variability of the separation factor indicates that while treating lower concentrations, the high value of R_L_ implies a reduced affinity adsorbate–adsorbent. On the other hand, the treatment of concentrated solutions showed a reduction in the R_L_ values close to 0, strengthening the chromium attachment onto the CNF surface [83].

The mean free energy of adsorption calculated through the Dubinin–Raduskevich model was 2.24 kJ·mol^−1^, similar to the typical values indicated for the physisorption of chromium (<8 kJ·mol^−1^) [84]. The value of the Temkin b_T_ parameter, which is related to the heat of sorption, was 0.19 kJ·mol^−1^. Choudhary and Paul [85] indicated that values of b_T_ below 8 kJ·mol^−1^ revealed a weak interaction chromium-CNF surface. These low b_T_ values are related to physisorption processes, where the values of adsorption enthalpy are in the order of physical processes such as intermolecular forces.

## 4. Conclusions

The modified CNF hydrogels successfully removed more than 97% of Cr(VI), opening promising applications for these nanomaterials as heavy metal adsorbents. The nanofibers showed a large amount of active carboxylic and anionic groups. Thus, it was necessary to develop a surface modification through hydrophobization treatment. The inclusion of 1.5 mmol MTMS·g^−1^ as a hydrophobizing agent allowed for the increase in the Cr(VI) kinetic constant of adsorption k_2_ by 84.97%. The pseudo-second order and intraparticle diffusion kinetic models were the best fitting models. These models revealed that both the sorption rate and hexavalent chromium diffusion played a major role as rate-limiting steps. The adsorption mechanism is ruled by the multi-step adsorption of hexavalent chromium on the CNF hydrogel dominated by internal diffusion at low concentrations and external diffusion with concentrations above 5 mg·L^−1^. The optimized conditions were found to be pH 3 and a dosage over 500 mg·L^−1^. More than 97% of hexavalent chromium removal was reached, treating concentrations below 1 mg·L^−1^ and the maximum adsorption capacity of 70.38 mg·g^−1^ was achieved at 50 mg·L^−1^. The isotherm analysis showed that Freundlich was the best fitting model, meaning that multilayer adsorption and the heterogeneous dispersion of surface energy is the main adsorption mechanism of hexavalent chromium onto the surface of hydrophobized CNF. The Freundlich unfavorable isotherm predicts a multilayer adsorption and a weak interaction between hexavalent chromium and CNF, associated with a physical sorption mechanism. The relatively low values of mean free energy of adsorption (2.24 kJ·mol^−1^) and heat of sorption (0.19 kJ·mol^−1^) calculated through the Dubinin–Raduskevich and Temkin models are also indicators of a physical sorption process. In general terms, the hydrophobized CNF hydrogel reached relatively high adsorption capacities compared to previously developed cellulose nanomaterials and activated carbons, and the complete removal of chromium could be found, even at low adsorbent dosages. 

## Figures and Tables

**Figure 1 polymers-14-03425-f001:**
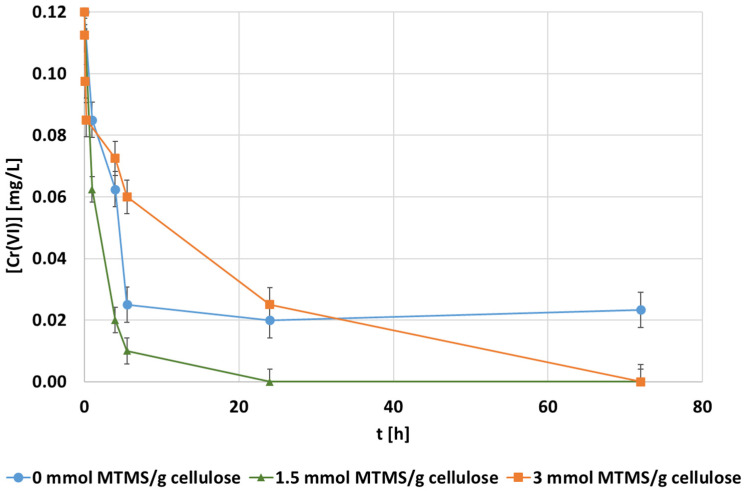
The evolution of the hexavalent chromium concentration [mg·L^−1^] at 0.1 mg·L^−1^ of the initial chromium concentration, pH 3, and 1000 mg CNF·L^−1^ of dosage during adsorption with different doses of MTMS in the CNF hydrogels.

**Figure 2 polymers-14-03425-f002:**
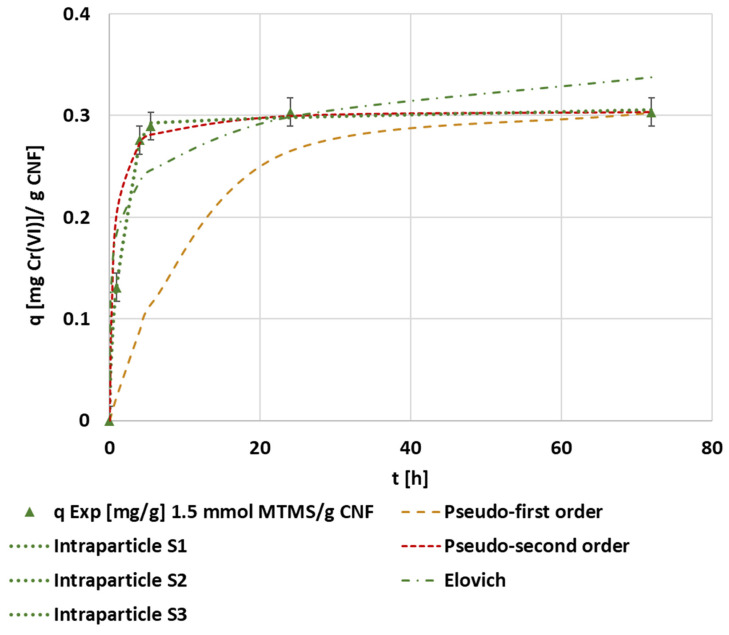
The evolution of the kinetic adsorption experiment at 0.1 mg·L^−1^ of the initial chromium concentration, pH 3, and 1000 mg CNF·L^−1^ hydrophobized with 1.5 mmol MTMS·g^−1^ cellulose and kinetic fitting of the pseudo-first, pseudo-second, Elovich, and intraparticle models.

**Figure 3 polymers-14-03425-f003:**
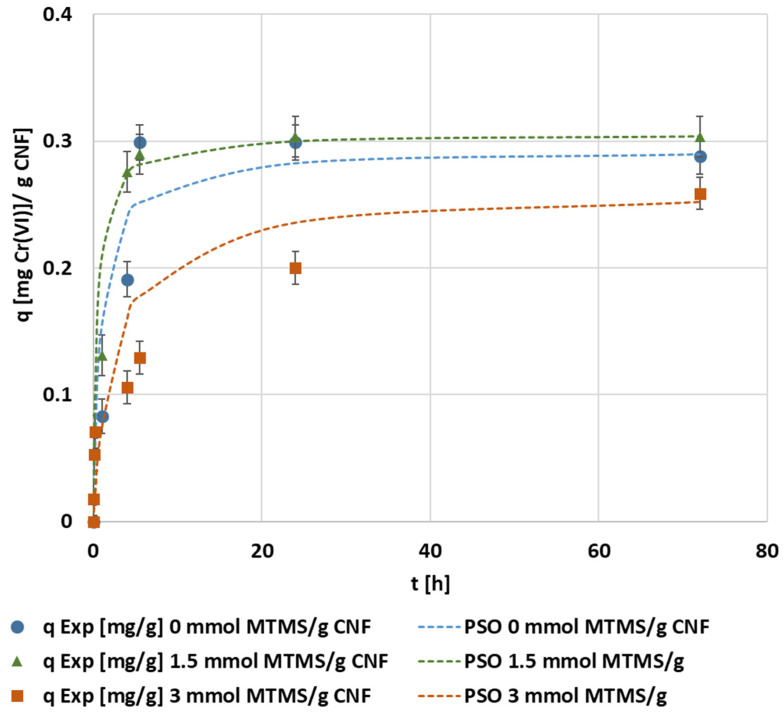
The evolution of the kinetic adsorption experiment at the previously indicated conditions under different MTMS dosages and kinetic fitting of the pseudo-second order model.

**Figure 4 polymers-14-03425-f004:**
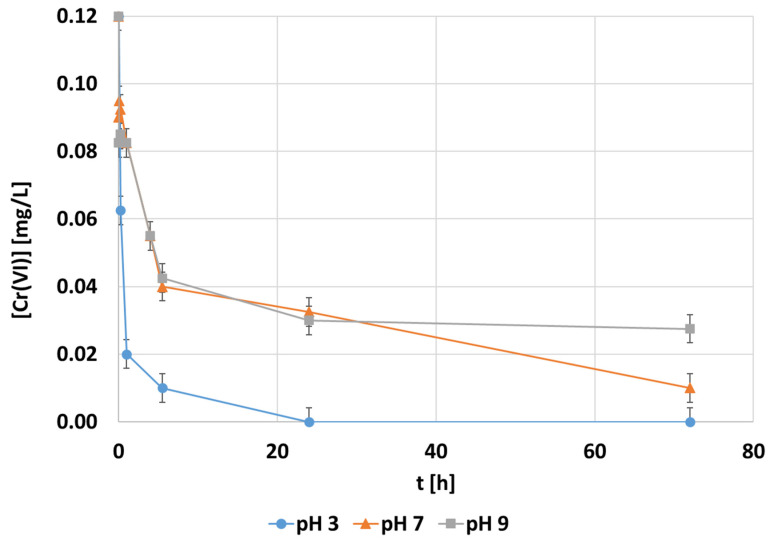
The evolution of the hexavalent chromium concentration [mg·L^−1^] during adsorption with the CNF hydrogel at 0.1 mg·L^−1^ of the initial chromium concentration, 1.5 mmol MTMS·g^−1^ CNF applied during hydrophobization, and 1000 mg CNF·L^−1^ of dosage under different pH conditions.

**Figure 5 polymers-14-03425-f005:**
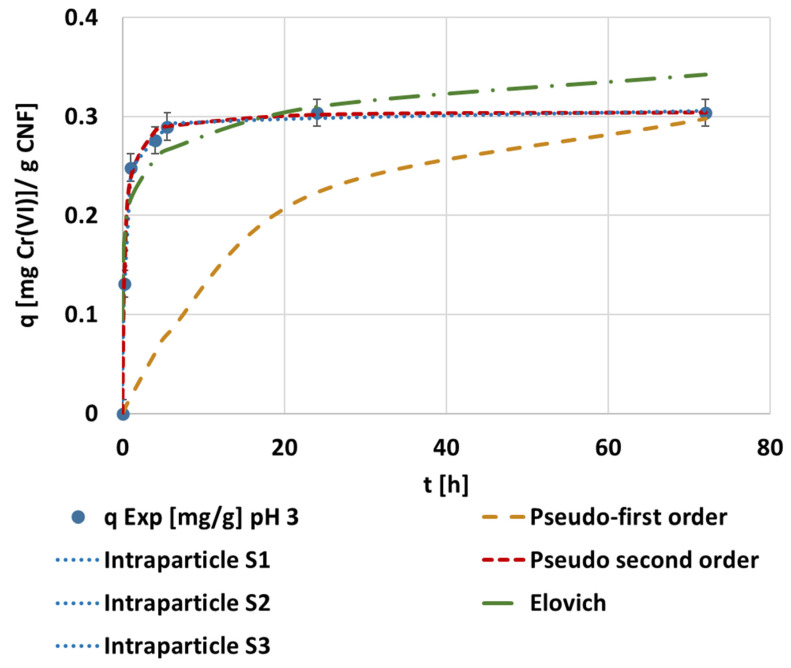
The evolution of the kinetic adsorption experiment at 0.1 mg·L^−1^ of the initial chromium concentration, 1.5 mmol MTMS·g^−1^ CNF applied during hydrophobization and 1000 mg CNF·L^−1^ of dosage under pH 3 conditions and the kinetic fitting of the pseudo-first, pseudo-second, Elovich, and intraparticle models.

**Figure 6 polymers-14-03425-f006:**
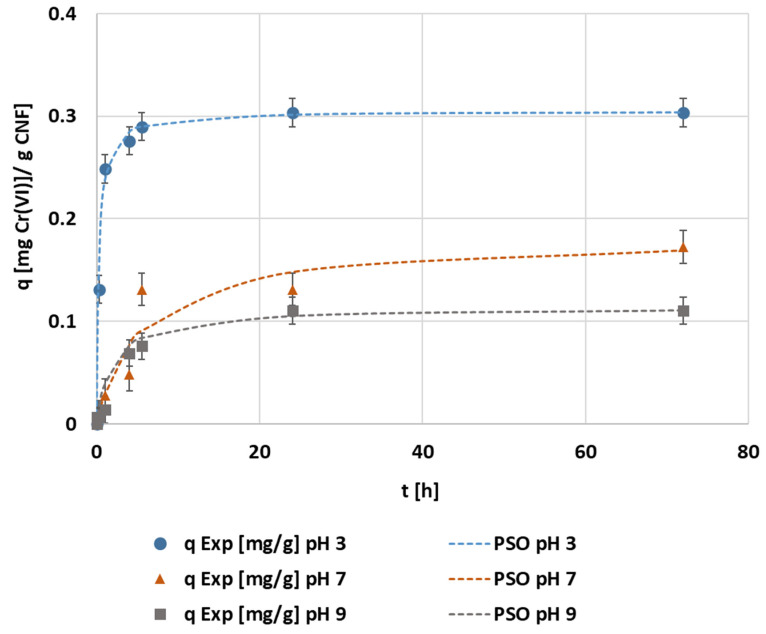
The evolution of the kinetic adsorption experiment at 0.1 mg·L^−1^ of the initial chromium concentration, 1.5 mmol MTMS·g^−1^ CNF applied during hydrophobization, and 1000 mg CNF·L^−1^ of dosage under different pH and kinetic fitting of the pseudo-second order model.

**Figure 7 polymers-14-03425-f007:**
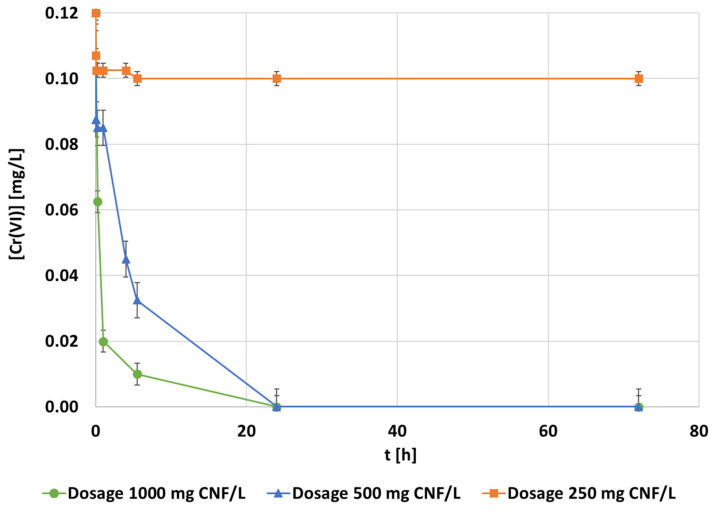
The evolution of the hexavalent chromium concentration [mg·L^−1^] during adsorption with the CNF hydrogel at 0.1 mg·L^−1^ of the chromium initial concentration, 1.5 mmol MTMS·g^−1^ CNF, and pH 3 conditions under different adsorbent dosages.

**Figure 8 polymers-14-03425-f008:**
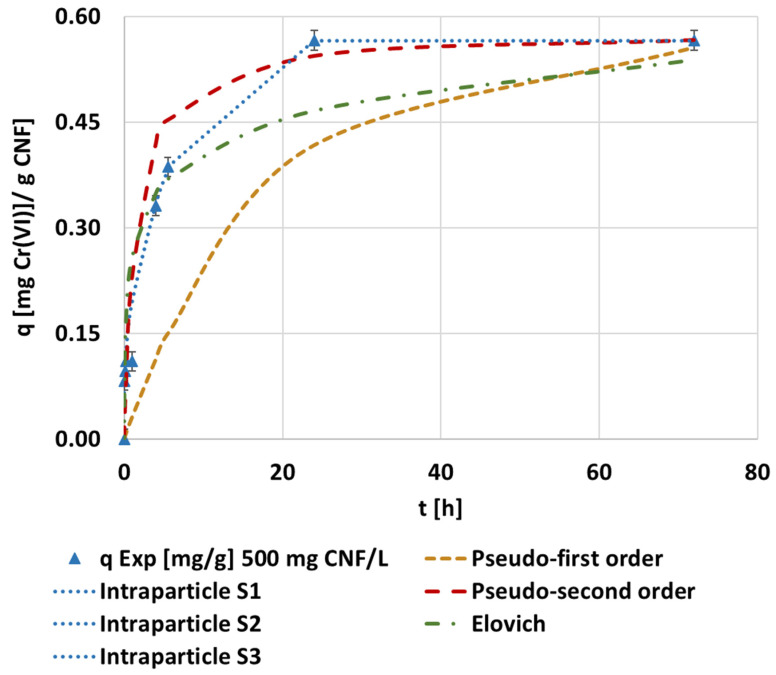
The evolution of the kinetic adsorption experiment at 0.1 mg·L^−1^ of the chromium initial concentration, 1.5 mmol MTMS·g^−1^ CNF, and pH 3 conditions under 500 mg CNF·L^−1^ of dosage and the kinetic fitting of the pseudo-first, pseudo-second, Elovich, and intraparticle models.

**Figure 9 polymers-14-03425-f009:**
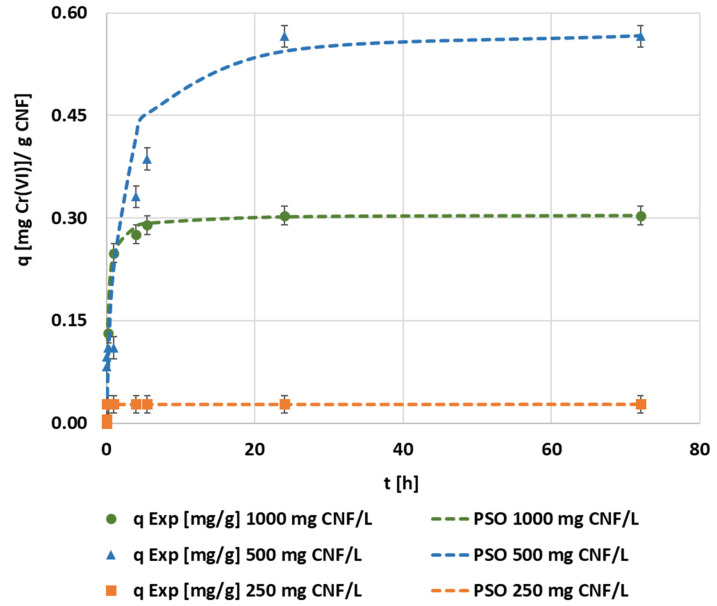
The evolution of the kinetic adsorption experiment at 0.1 mg·L^−1^ of the chromium initial concentration, 1.5 mmol MTMS·g^−1^ CNF, and pH 3 conditions under different adsorbent dosages and kinetic fitting of the pseudo-second order model.

**Figure 10 polymers-14-03425-f010:**
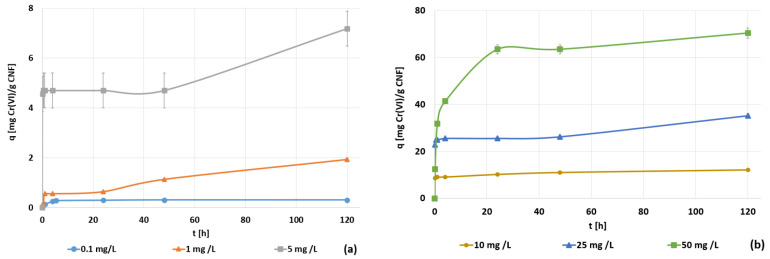
(**a**) The evolution of the hexavalent chromium concentration [mg·L^−1^] during the adsorption with CNF hydrogel at 1000 mg·L^−1^ of dosage, 1.5 mmol MTMS·g^−1^ CNF, and pH 3 conditions under 0.1 to 5 mg·L^−1^; (**b**) 10 to 50 mg·L^−1^ initial hexavalent chromium concentrations.

**Figure 11 polymers-14-03425-f011:**
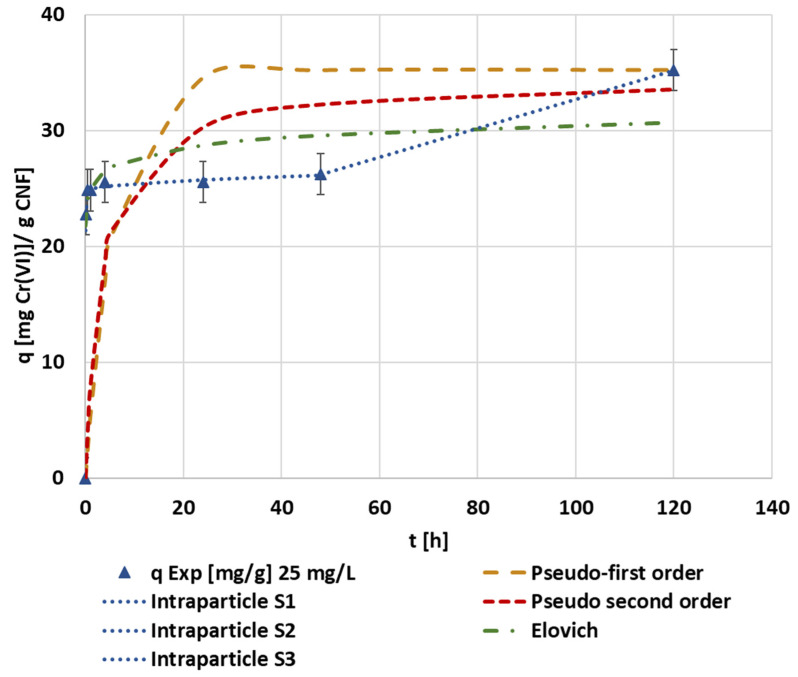
The evolution of the kinetic adsorption experiment of the CNF hydrogel at 1000 mg·L^−1^ of dosage, 1.5 mmol MTMS·g^−1^ CNF, and pH 3 conditions under 25 mg·L^−1^ of the initial hexavalent chromium concentration and kinetic fitting of the pseudo-first, pseudo-second, Elovich, and intraparticle models.

**Figure 12 polymers-14-03425-f012:**
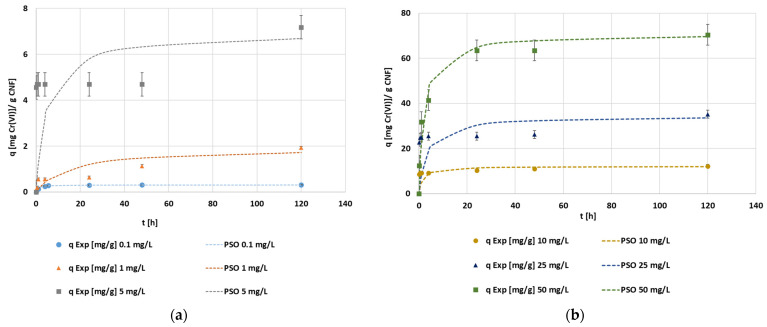
(**a**) The evolution of the kinetic adsorption experiment of the CNF hydrogel at 1000 mg·L^−1^ of dosage, 1.5 mmol MTMS·g^−1^ CNF, and pH 3 conditions under 0.1 to 5 mg·L^−1^; (**b**) 10 to 50 mg·L^−1^ of the initial hexavalent chromium concentration and kinetic fitting of the pseudo-second order model.

**Table 1 polymers-14-03425-t001:** The nonlinear and linearized equations of the analyzed kinetic models.

Model	Nonlinearized Equations		Linearization		Ref.
Pseudo-first order ^1^	$q=q_{e}\cdot \left(1-e^{-k_{1}\cdot t}\right)$	(4)	$\ln\left(q_{e}-q\right)-\ln\left(q_{e}\right)=-k_{1}\cdot t$	(8)	[43]
Pseudo-second order	$q=\frac{q_{e}^{2}\cdot k_{2}\cdot t}{1+q_{e}\cdot k_{2}\cdot t}$	(5)	$\frac{t}{q}=\left(\frac{1}{q_{e}^{2}\cdot k_{2}}\right)+\left(\frac{1}{q_{e}}\right)\cdot t$	(9)	[43]
			$\frac{1}{q}=\left(\frac{1}{q_{e}^{2}\cdot k_{2}}\right)\cdot \frac{1}{t}+\left(\frac{1}{q_{e}}\right)$	(10)	[44]
Elovich	$q=\frac{1}{\beta }\cdot \ln\left(t\right)+\frac{1}{\beta }\ln\left(\alpha \cdot \beta \right)$	(6)	-		[45]
Weber and Morris (Intraparticle)	$q=k_{i}\cdot t^{0.5}+C$	(7)	-		[46]

^1^ The value of q_e_ must be previously obtained by estimation or experimentally.

**Table 2 polymers-14-03425-t002:** The nonlinear and linearized equations of the analyzed isotherm models.

Model	Nonlinearized Equations		Linearization		Ref.
Langmuir	$q_{e}=\frac{K_{L}\cdot q_{\max}\cdot C_{e}}{1+K_{L}\cdot C_{e}}$	(11)	$\mathrm{Type}\;I:\;\frac{C_{e}}{q_{e}}=\frac{1}{K_{L}\cdot q_{\max}}+\left(\frac{1}{q_{\max}}\right)\cdot C_{e}$	(20)	[47]
			$\mathrm{Type}\;\mathrm{II}:\;\frac{1}{q_{e}}=\frac{1}{q_{\max}}+\left(\frac{1}{K_{L}\cdot q_{\max}}\right)\cdot \left(\frac{1}{C_{e}}\right)$	(21)	
			$\mathrm{Type}\;\mathrm{III}:\;q_{e}=\left(-\frac{1}{K_{L}}\right)\cdot \left(\frac{q_{e}}{C_{e}}\right)+q_{\max}$	(22)	
			$\mathrm{Type}\;\mathrm{IV}:\;\frac{q_{e}}{C_{e}}=\left(-K_{L}\right)\cdot q_{e}+K_{L}\cdot q_{\max}$	(23)	
	$R_{L}=\frac{1}{1+K_{L}\cdot C_{0}}$	(12)	$\mathrm{Type}\;V:\;\frac{1}{C_{e}}=\left(K_{L}\cdot q_{\max}\right)\cdot \frac{1}{q_{e}}-K_{L}$	(24)	
Freundlich	$q_{e}=K_{F}\cdot C_{e}^{1/n_{F}}$	(13)	$\ln\left(q_{e}\right)=\frac{1}{n_{F}}\cdot \ln\left(C_{e}\right)+\ln\left(K_{F}\right)$	(25)	
Temkin	$q_{e}=B_{T}\cdot \ln\left(A_{T}\cdot C_{e}\right)$	(14)	$q_{e}=B_{T}\cdot \ln\left(A_{T}\right)+B_{T}\cdot \ln\left(C_{e}\right)$	(26)	
	$B_{T}=\frac{R\cdot T}{b_{T}}$	(15)			
Dubinin–Raduskevich	$q_{e}=q_{\max}\cdot \exp\left(-B_{\mathrm{DR}}\cdot \varepsilon^{2}\right)$	(16)	$\ln\left(q_{e}\right)=\ln\left(q_{\max}\right)-B_{\mathrm{DR}}\cdot \varepsilon^{2}$	(27)	
	$\varepsilon =R\cdot T\cdot \ln\left(1+\frac{1}{C_{e}}\right)$	(17)			
	$E=\frac{1}{\sqrt{2\cdot B_{\mathrm{DR}}}}$	(18)			
Sips ^1^	$q_{e}=\frac{K_{S}\cdot q_{\max}\cdot C_{e}^{1/n_{S}}}{1+K_{S}\cdot C_{e}^{1/n_{S}}}$	(19)	$\ln\left(\frac{q_{e}}{q_{\max}-q_{e}}\right)=\frac{1}{n_{s}}\cdot \ln\left(C_{e}\right)+\ln\left(K_{S}\right)$	(28)	

^1^ The value of q_max_ can be first estimated from the q_max_ obtained from the Langmuir model as the first input for optimization using a calculation software.

**Table 3 polymers-14-03425-t003:** A comparison of the hexavalent chromium adsorption through different NC and activated carbon adsorbents.

Adsorbent	Contact Time [min]	Adsorbent Dosage [mg·L^−1^]	Initial Cr(VI) Concentration [mg·L^−1^]	pH	q_max_ [mg·g^−1^]	Maximum Removal Yield [%]	Ref.
CNF from rice husk	100	1500	30	6	3.76	92.99	[57]
Polypyrrole-bacterial CNF	180	250	300	2	555.6	97.5	[72]
Thiol-modified CNF composite	20		50	4	87.5	96	[71]
Citric acid-incorporated CNF	120	40	50	2	11	23	[77]
Amino-silanized cellulose membranes	300	5000	50	4	34.7		[33]
Polyaniline-functionalized CNC	40	500	30	2.5	48.92	97.84	[73]
Microwave-assisted H_3_PO_4_/Fe-modified activated carbon	200	1000	30	3	34.39	100	[74]
ZnCl_2_-modified tamarind wood activated carbon	70	3000	10	3	28.02	99	[75]
Acid-base surface modified activated carbon	180	2000	50		13.89		[76]
Hydrophobized CNF Hydrogel (MTMS dosage = 1.5 mmol·g^−1^)	330	500	50	3	70.38	>97.14	This work

## Data Availability

Not applicable.

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
