# Peer review of "Modeling of Hexavalent Chromium Removal with Hydrophobically Modified Cellulose Nanofibers"

_polymers, 2022, doi:10.3390/polym14163425_

Round 1

Reviewer 1 Report

Manuscript Title: Modeling of Hexavalent Chromium Removal with Hydrophobically Modified Cellulose Nanofibers

Reviewer Comments:

General comment:

This paper is regarding a research on modeling of hexavalent chromium removal with hydrophobically modified cellulose nanofibers. The script in this current form can be revised to achieve publication quality. There are some clarifications needed to understand the processes carried out in this work. To conclude, this paper needs to be revised carefully before it can be considered in journal like Polymers. Hope that comments below will be able to help to further improve the paper

    Please check Guides for Authors to make sure it is followed strictly

    Language: There are some language errors (tenses, singular/plural) and incomplete sentences in the script. Please check the sentence structure, tenses and language carefully in the revised manuscript.

    Standardize the unit convention.

Abstract:

    Needs minor revisions prior to the amendment of the main content.

    An abstract is often presented separately from the article, so it must be able to stand alone. Hence the problem statement, aim, novelty and results of the study have to be included in.

Introduction:

    Describe more on the environmental and health issues together with treatment possibility and methods

    Further highlight novelty in last paragraph and the work carried out in this paper

    Kindly refer papers below as they are highly relevant to this report:

    “Biosorption potential of Phoenix dactylifera coir wastes for toxic hexavalent chromium sequestration”

    “Cellulose acetate-based membranes by interfacial engineering and integration of ZIF-62 glass nanoparticles for CO2 separation”

    “Natural hydroxyapatite from fishbone waste for the rapid adsorption of heavy metals of aqueous effluent”

    “Removal of calcium ions from aqueous solution by bovine serum albumin (BSA)-modified nanofiber membrane: Dynamic adsorption performance and breakthrough analysis”

Main body:

    The main objective and novelty of this work is still deemed not highlighted enough. The authors should put in more efforts to revise the discussion properly in order to let the readers understand the importance of this work.

    Kindly improve on the discussion. What is the significance of the results of the work? Include more relevant literatures.

    Further enhance the discussion section, together with the results.

Conclusion

    Kindly improve to include in more concise and significant results.

    Should include some present challenges and possible routes to improve them. Describe them in more details.

Author Response

General comment:

This paper is regarding a research on modeling of hexavalent chromium removal with hydrophobically modified cellulose nanofibers. The script in this current form can be revised to achieve publication quality. There are some clarifications needed to understand the processes carried out in this work. To conclude, this paper needs to be revised carefully before it can be considered in journal like Polymers. Hope that comments below will be able to help to further improve the paper

  • Please check Guides for Authors to make sure it is followed strictly.
  • Language: There are some language errors (tenses, singular/plural) and incomplete sentences in the script. Please check the sentence structure, tenses and language carefully in the revised manuscript.
  • Standardize the unit convention.

The revised manuscript has been carefully revised before resubmission. All comments from reviewer 1 have been thoroughly checked and considered to improve the manuscript. Some parts of the manuscript have been rewritten to clarify the text.

Abstract:

  • Needs minor revisions prior to the amendment of the main content.
  • An abstract is often presented separately from the article, so it must be able to stand alone. Hence the problem statement, aim, novelty and results of the study have to be included in.

Abstract has been rewritten as suggested including the problem, aim, novelty and main results to improve its quality and be easier to follow without the complete document.

Introduction:

  • Describe more on the environmental and health issues together with treatment possibility and methods.
  • Further highlight novelty in last paragraph and the work carried out in this paper
  • Kindly refer papers below as they are highly relevant to this report:
  • “Biosorption potential of Phoenix dactylifera coir wastes for toxic hexavalent chromium sequestration”
  • “Cellulose acetate-based membranes by interfacial engineering and integration of ZIF-62 glass nanoparticles for CO2 separation”
  • “Natural hydroxyapatite from fishbone waste for the rapid adsorption of heavy metals of aqueous effluent”
  • “Removal of calcium ions from aqueous solution by bovine serum albumin (BSA)-modified nanofiber membrane: Dynamic adsorption performance and breakthrough analysis”

Following the indications the introduction has been improved with further information on environmental and health issues (Lines 39-44); and treatment possibilities and methods (Lines 52-55, 58-71 and 74-75). The last paragraph has been rewritten to highlight the novelty and the major role of this work for further implementation of this treatment (Lines 120-124 and 128-129).

Main body:

  • The main objective and novelty of this work is still deemed not highlighted enough. The authors should put in more efforts to revise the discussion properly in order to let the readers understand the importance of this work.
  • Kindly improve on the discussion. What is the significance of the results of the work? Include more relevant literatures.
  • Further enhance the discussion section, together with the results.

The authors have remarked the relevance of the work. Further relevant papers in the field has been included to facilitate the assessment of the better performance of the proposed adsorbents with respect to other materials considering the chromium removal yield and the adsorption capacities. The authors have integrated the discussion section with the results to make the paper easier to read and follow. Results obtained are compared with other types of adsorbents (cellulosic and non-cellulosic ones), specially to understand the relevance of the low amount of optimized adsorbent dosage obtained in this work to adsorb hexavalent chromium, which opens the way for its future implementation (lines 234-238, 242-243, 245-246, 253-262, 286- 290, 328-333, 379-381, 396-405, 418-421, 496-512).

Conclusion

  • Kindly improve to include in more concise and significant results.
  • Should include some present challenges and possible routes to improve them. Describe them in more details.

Following the indications of the reviewer 1, the conclusion of the paper includes the major results of the developed work, i.e. optimal operation conditions achieved in the experiments, the maximum adsorption capacity, the maximum hexavalent chromium removal, the best fitting isotherm and kinetic models, as well as the explanation of the mechanism associated to these models and the obtained thermodynamic parameters of the process. Furthermore, the improvements reached by applying CNFs compared to other materials in hexavalent chromium adsorption has been also remarked.

Reviewer 2 Report

This article is devoted to the study of the adsorption of hexavalent chromium by hydrophobic cellulose derivatives. The article is well structured and written in clear language. The subject of this study is relevant. The paper considers various models of adsorption. There are some points that would like to be improved:

1. Unification of all drawings is desirable. Bring them to a common size and text size.

2. If all drawings are with colored lines, then this should be preserved. Please pay attention to pictures 2,5,8,11.

3. It is desirable to add more comparison with experimental data from other studies.

4. More references to the literature can be added in the discussion of the results.

5. Please add a reference to the literature, cite: 10.1021/acsomega.1c02570.

Author Response

This article is devoted to the study of the adsorption of hexavalent chromium by hydrophobic cellulose derivatives. The article is well structured and written in clear language. The subject of this study is relevant. The paper considers various models of adsorption. There are some points that would like to be improved:

  1. Unification of all drawings is desirable. Bring them to a common size and text size.

Thank you for the comment. The figures have been modified following the indications, fonts, font sizes and styles indicated in the guide for authors. Some of them show small size but are plotted in the document following the indications of Polymers’ template for authors for figures with multiple panels (figure 2 of the template).

  1. If all drawings are with colored lines, then this should be preserved. Please pay attention to pictures 2,5,8,11.

According to the suggestion, Figures 2, 5, 8, 11 have been modified to show a single color each curve.

  1. It is desirable to add more comparison with experimental data from other studies.

Data discussion has been improved including more references to compare our data with other types of adsorbents (cellulosic and non-cellulosic ones), specially to understand the relevance of the low value of optimized adsorbent dosage obtained in the work, which opens the way for its future implementation (lines 234-238, 242-243, 245-246, 253-262, 286- 290, 328-333, 379-381, 396-405, 418-421, 496-512).

  1. More references to the literature can be added in the discussion of the results.

As already indicated more references have been included to improve the data discussion and to highlight the relevance of our results. Data are compared with published results of kinetic studies and chromium removal yield and adsorption capacities with different adsorbents (lines 234-238, 242-243, 245-246, 253-262, 286- 290, 328-333, 379-381, 396-405, 418-421, 496-512).

  1. Please add a reference to the literature, cite: 10.1021/acsomega.1c02570.

The mentioned reference has been included in the article (Ref. 20).